# Diagnosing and engineering gut microbiomes

Elisa Cappio Barazzone[1,2,5], Médéric Diard[2,3,5], Isabelle Hug [2,3,5], Louise Larsson [1,2,5] & Emma Slack [1,2,4,5 ✉]

## Abstract

The microbes, nutrients and toxins that we are exposed to can have a profound effect on the composition and function of the gut microbiome. Thousands of peer-reviewed publications link microbiome composition and function to health from the moment of birth, right through to centenarians, generating a tantalizing glimpse of what might be possible if we could intervene rationally. Nevertheless, there remain relatively few real-world examples where successful microbiome engineering leads to beneficial health effects. Here we aim to provide a framework for the progress needed to turn gut microbiome engineering from a trial-and-error approach to a rational medical intervention. The workflow starts with truly understanding and accurately diagnosing the problems that we are trying to fix, before moving on to developing technologies that can achieve the desired changes.

**Keywords** Microbiota Engineering; Gut; Bacteriophage; Probiotic; Vaccination

**Subject Categories** Immunology; Microbiology, Virology & Host Pathogen Interaction

## Introduction

From the moment of birth, mammals inhabit a microbial world. Our first exposure is usually to maternal microbes ingested during birth. The maturing microbiome then rapidly changes during the first years of life, reaching an adult-like composition at around the age of three years in humans (Yatsunenko et al, 2012). All body surfaces become colonized during this phase of life, and the importance of the microbiome on the skin and in the respiratory and urogenital tracts has been recently reviewed elsewhere (Flynn et al, 2022; Drigot and Clark, 2024; France et al, 2022; Harris-Tryon and Grice, 2022). For the purpose of this review, we will focus on the gut (Fig. 1). *Enterobacteriaceae* are abundant in the gut microbiome during the first days of life, then microbial diversity gradually increases with an enrichment in phyla *Bacillota* and *Bacteroidota* (Yatsunenko et al, 2012). *Bifidobacteriaceae*, which have been associated with beneficial infant microbiome

compositions, colonize faster in infants born vaginally compared to infants born via C-section (Duranti et al, 2017; Reyman et al, 2019; Mitchell et al, 2020). Breast milk further supports the growth of *Bifidobacteriaceae* strains that are able to use human milk oligosaccharides (HMOs) and urea (Lawson et al, 2020).

The transition to solid food correlates with the maturation of the microbiome toward an adult-like state with abundant short-chain fatty acids (SCFA) producers. The microbiome in adults comprises around $10^{13}$ bacteria from 250 to 1000 species as well as archea, fungi, viruses, protozoa and multicellular eukaryotes (Beller et al, 2021; Gilbert et al, 2018) (Fig. 1). Overall phyla distribution in the adult human gut microbiome is remarkably conserved and typically dominated by *Bacillota* and *Bacteroidota*. Similar host-phylum-specific microbiota patterns can be observed in all mammalian species studied so far (Muegge et al, 2011), implying host control of the gut microbiome composition. Some uniformity is also observed at the level of core microbial metabolites produced by the microbiota across individuals. Nevertheless, when one looks in more detail, very extensive variation is seen in secondary metabolites and at the bacterial species and strain level, even between genetically identical hosts. This is likely related to diet and exposure to commensal and pathogenic microbes, medications, stochastic effects, as well as time. For example, *E. coli* strains seem to turn over in the human gut, on average, every few months (Worby et al, 2022; Martinson et al, 2019). Some strains show diurnal variations in abundance and most show diurnal variation in function (Thaiss et al, 2016; Hoces et al, 2022). The influence of these shifts and inter-individual differences on host health is a highly active area of research.

## Diagnosing the microbiome: what are we trying to fix?

To engineer something effectively, one needs to understand how it works normally, and what is currently wrong with it. While we can observe changes in microbiome composition or metabolism between healthy controls and patients with a wide range of diseases, identifying species that may be causative remains challenging. Indeed, this may even be the wrong approach altogether, as the behaviour of microbial consortia can be strongly defined by species-species interactions, producing dynamics that are challenging to predict from first principles (Coyte et al, 2015; Weiss et al, 2023, 2022). As a way of dealing with this complexity,

[1]Laboratory for Mucosal Immunology, Institute for Food, Nutrition and Health, Department of Health Sciences and Technology, ETH Zurich, Zürich, Switzerland. [2]Basel Research Centre for Child Health, Basel, Switzerland. [3]Biozentrum, University of Basel, Basel, Switzerland. [4]Sir William Dunn School of Pathology, University of Oxford, Oxford, UK. [5]These authors contributed equally: Elisa Cappio Barazzone, Médéric Diard, Isabelle Hug, Louise Larsson, Emma Slack. ✉E-mail: emma.slack@hest.ethz.ch

### Glossary

**Bacillota (formerly Firmicutes)**
A phylum of bacteria commonly found in the human gut, known for their role in the digestion of complex carbohydrates and production of short-chain fatty acids (SCFAs).

**Bacteroidota (formerly Bacteroidetes)**
A phylum of bacteria that are major components of the gut microbiota, involved in the breakdown of complex molecules such as polysaccharides and proteins.

**Bacteriophage (Phage)**
A virus that infects bacteria, and can be used as a tool to specifically target and eliminate bacterial strains.

**Bifidobacteriaceae**
A family of bacteria often dominant in the gut microbiome of infants, particularly those who are breastfed, known for their role in digesting human milk oligosaccharides (HMOs).

**Capsular polysaccharides**
Complex sugar molecules that form a protective layer around some bacteria, influencing their ability to evade the immune system and colonize the host.

**Colonization resistance**
The ability of the native gut microbiome to prevent colonization of pathogenic organisms through competition for nutrients and space or the production of inhibitory substances.

**Commensal microbes**
Microorganisms that live in a complex relationship with their host, contributing to essential physiological processes as well as disease.

**Competitive exclusion**
A principle in ecology stating that two species competing for the same resources cannot coexist at constant population values if other ecological factors remain constant. In the context of microbiomes, it refers to the idea that beneficial bacteria can prevent the growth of harmful bacteria by outcompeting them for nutrients and space.

**CRISPR**
Acronym for "Clustered Regularly Interspaced Short Palindromic Repeats". CRISPR is a natural part of prokaryotic adaptive immunity and has been developed into a revolutionary genetic engineering technology that allows for precise editing of DNA in organisms.

**Dysbiosis**
A term used to describe an imbalance or maladaptation of the microbiome, often associated with disease. Dysbiosis can involve a loss of microbial diversity, an overgrowth of pathogenic microbes, or a decline in beneficial microbial functions.

**Engraftment**
The process of establishing and maintaining a new microorganism (or a consortium of microorganisms) in an existing microbial community.

**Enterobacteriaceae**
A family of gamma proteobacteria commonly found in the gut, especially in the early days of life, which includes both commensal and pathogenic species, such as *Klebsiella* and *Escherichia coli*.

**Human milk oligosaccharides (HMOs)**
Complex carbohydrates found in human breast milk that promote the growth of beneficial gut bacteria, particularly *Bifidobacteriaceae*, in infants.

**Hyperammonemia**
A condition characterized by elevated levels of ammonia in the blood, often due to liver dysfunction. Engineered bacteria may be used to reduce ammonia levels by converting it into less harmful compounds.

**Metabolomics**
The study of small molecules, within cells, fluids, tissues, or organisms. Metabolomics provides a snapshot of the metabolic state and offers insights into different physiological and pathological conditions.

**Niche**
A specific role or position a (micro)organism occupies within an ecosystem, including the resources it uses and the interactions it has with other organisms.

**Phage steering**
A strategy that uses bacteriophages to exert selective pressure on bacteria, leading to the evolution of less harmful or more treatable bacterial strains.

the term "dysbiosis" has emerged as a very broad way of describing the microbiota in various disease states. It should be noted that the term itself remains something of a circular argument as "microbiotas found in cases of disease" are not per se dysfunctional. Attempts to physically define dysbiosis typically revolve around loss of diversity, loss of specific functions or overrepresentation of facultative anaerobes (Johnson and Burnet, 2016; Lozupone et al, 2012). However, none of these features reliably predict disease versus health except in very extreme cases, such as vancomycin-resistant *Enterococcus* overgrowth in bone-marrow transplant patients, or very dramatic *Enterobacteriaceae* or *Clostridioides difficile* overgrowth (Tavadze et al, 2014). Moreover, there are cases, such as the inborn errors of metabolism affecting odd-chain fatty acids or ammonia metabolism, where it is the functions of a healthy microbiome that are thought to be disease driving (Helman et al, 2014; Fowler et al, 2008). Furthermore, alterations in host physiology during disease generally drive a shift in microbiome composition, which complicates interventions solely addressing the microbial shifts without addressing the causative physiological change. We therefore use the term with caution here. If we want rational microbiome interventions to work, we need to go beyond "dysbiosis" and precisely, molecularly, define what is wrong and why.

## What tools do we have to diagnose the microbiome?

Microbiome science is only as good as our ability to measure the relevant parameters that describe microbiome composition and function. Correspondingly, technological progress continues to be the main driver of our understanding. Prior to the year 2000, we were limited by our ability to detect and quantify individual species beyond those that could be isolated and cultivated. With the broad availability of high-throughput sequencing, 16S rDNA amplicon sequencing and metagenomics have become the dominant analysis tools (Yi et al, 2024). Insights range from a broad descriptions of phylum- and genus-level composition of the gut microbiome, down to the variation in the abundance of metabolic pathways or of the individual strains associated with disease states. In parallel, there is major progress in populating databases with data linking species, genes and functions, and in bioinformatic tools that allow us to extract meaning. The potential power of this approach is impressive and mirrored by the amount of financial and political weight behind missions such as the Integrative Human Microbiome Project (Proctor et al, 2019).

This said, we need to be aware of the limitations of high-throughput sequencing approaches.

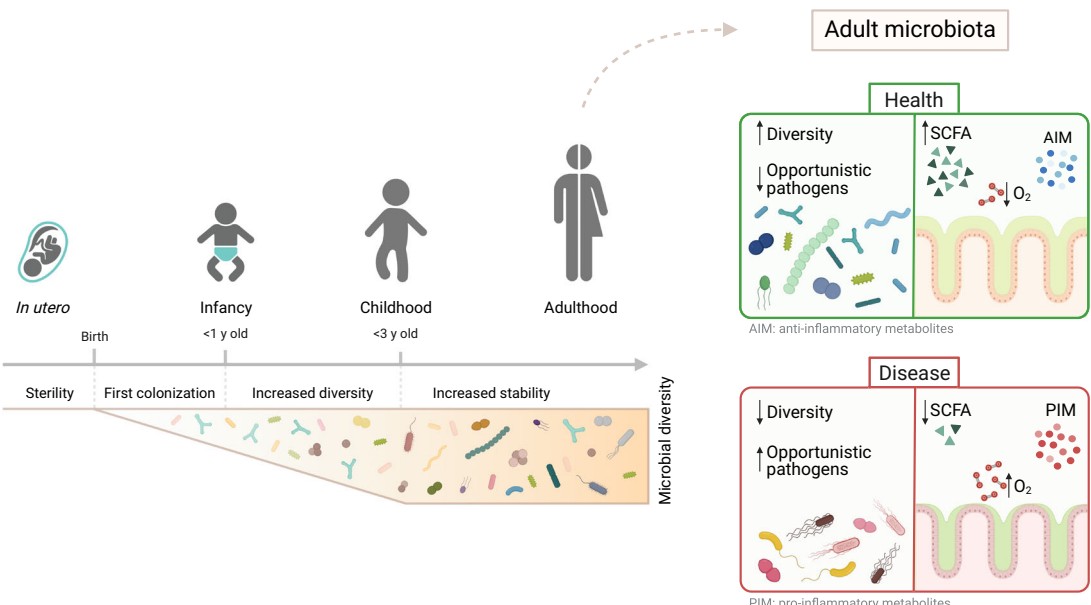

**Figure 1. Microbiome complexity increases during early life.**

The number of species and typical species distribution reaches adult-like levels from the age of around 3 years. After this stage, there has been an association of a range of diseases with decreased microbiome diversity and higher densities of opportunistic pathogens. Graphics were created with BioRender.com.

(1) Low-abundance species: The long-tail of low-abundance species—typically anything present at a density of below $10^8$ bacteria per gram in the large intestinal microbiota—are not reliably detected (Reitmeier et al, 2021). These species are sometimes dismissed as too rare to be functionally relevant. However, this density is more than sufficient for *Salmonella* or a pathogenic *E. coli* to drive overt disease (World Health Organization & Food and Agriculture Organization of the United Nations, 2002; Gopinath et al, 2012), suggesting we may be overlooking critical functions.

(2) Technical variation: DNA extraction methods, primers selection and quality control, curation of databanks, and analysis algorithms can cause huge variation between datasets (Shaffer et al, 2022; Reitmeier et al, 2021). For low-biomass samples, physical contamination is also a major issue. This has led to confusion in sterile tissues such as the placenta, which can nevertheless generate sequencing reads that map to bacterial genomes (Kennedy et al, 2023; Salter et al, 2014).

(3) "Dark matter": New metabolic pathways are still being uncovered in *E. coli* K12, which is one of the best-studied microbes on the planet (Hanson et al, 2021; Denger et al, 2014). In addition, many proteins show homology, yet perform differing functions, while unrelated proteins can perform the same activity (Pearson, 2015). Therefore, going from metagenome-assembled genomes to genome-scale metabolic maps generates useful information, but is not currently likely to be the whole truth.

(4) Extrachromosomal DNA may encode for critical functions, but is extremely challenging to correctly assemble and assign to a host strain, or may be missed altogether in metagenomic analysis (Antipov et al, 2019). Frequent horizontal gene transfer in densely populated environments such as the gut

also poses challenges for linking functions to species (Moura de Sousa et al, 2023).

Sequencing technologies continue to evolve rapidly. Recently, Long Read Sequencing (LRS) technologies have greatly increased our power to study complex microbial communities. LRS allows the assembly of genomes containing repeated sequences that Short Read Sequencing (SRS) cannot resolve. LRS can also detect genomic rearrangements that play an important role in microbial evolution (Bharti and Grimm, 2021). However, by implementing these sequencing techniques alone, mobile extrachromosomal elements, such as plasmids, cannot be assigned to their host strains, limiting our understanding of microbial communities' evolution by horizontal gene transfer. Through cross-linking chromosomes with extrachromosomal genetic material, chromosome conformation capture on metagenomes (meta3C) (Marbouty and Koszul, 2015) offers a promising solution to this issue. Moreover, an upgraded version of meta3C, namely metaHiC, can even be used to assign bacteriophages to their hosts and to better understand the impact of bacteriophage predation on the dynamics of complex bacterial populations (Marbouty et al, 2021).

Another major approach to diagnosing microbiomes is to try to directly measure microbiome functions, such as through metabolome assessment (Bauermeister et al, 2022). Analysis of easily accessible fluids such as urine, fecal water and blood allows quite frequent longitudinal sampling and useful insights into flux through pathways, particularly where isotope-labelled compounds are used (Berry and Loy, 2018; Berry et al, 2013). Techniques such as liquid- or gas-chromatography coupled to mass spectrometry or NMR-based approaches allow for the identification, and the quantification of microbial metabolites in these fluids. This enables broad surveys of low-molecular weight compounds with the potential to

reveal new targets for investigation. It is important to note that, the choice of metabolomics technique must align with the relevant compounds and samples, and the specific limitations of the chosen technique—including sample matrix, ionization efficiency, and ion suppression—should be well understood before interpreting the data (Ghosson et al, 2021; Hohenester et al, 2020; Lan et al, 2021).

The metabolome of the gut microbiota is highly circadian, meaning that both time- and health-dependence of the signals observed should be considered (Wang et al, 2017; Hoces et al, 2022; Thaiss et al, 2016). Besides, humans have a sigmoid colon that collects the fecal output over relatively long time periods. Time-dependent feces collection in humans is therefore non-trivial and invasive, while this can be relatively easily achieved in mice. In addition, feces represent only an end-product, as a large fraction of the gut microbiota metabolites are absorbed or further metabolized by the host higher up the intestine (Kozik et al, 2019; Moore et al, 2024). Swallowed robotic sampling devices offer a potential method to sample gut content from hard-to-access sites and verify variations in metabolites composition along the gastrointestinal tract (Shalon et al, 2023). An interesting alternative that has recently been explored is the quantification of metabolites in exhaled or ambient gases (Lan et al, 2021; García-Gómez et al, 2016; Singh et al, 2018; Wüthrich et al, 2022; Bruderer et al, 2020). We have recently demonstrated the direct detection of microbiome metabolites in the ambient gas around a live mouse using secondary electrospray ionization—high-resolution mass spectrometry (SESI-HRMS) (Lan et al, 2023). While this technique suffers from all of the challenges of direct injection mass spectrometry, it has the major advantage of allowing continuous non-invasive monitoring over a complete diurnal cycle. With improvements in soft ionization and compound identification, this could become a powerful technique to understand temporal variation in microbiota function.

Imaging techniques have also been developed that allow micrometre-scale localization of particular metabolites or metabolic functions in gut samples or in sections of the intestine. These include NanoSIMS (Nanoscale Secondary Ion Mass Spectrometry) as well as Raman-based imaging (Lee et al, 2020, 2021; Ge et al, 2022). As the healthy gut content is generally well-mixed (Arnoldini et al, 2018), spatial variation is typically only seen at microscopic scales, particularly very close to the epithelial barrier. However, in pathological conditions, changes in gut motility and secretions may dramatically alter this picture (McCallum and Tropini, 2024; Tropini, 2021). Raman spectroscopy has also been employed to sort microbes with specific metabolic functions, enabling genome-phenotype linkages in vivo (Lee et al, 2019).

Another class of microbiome diagnostic tools has recently emerged based on reporter bacterial strains (Tanna et al, 2021). Novel reporter techniques such as CRISPR-recording (Schmidt et al, 2022) and fluorescent reporter strains modified to record a "memory" of exposures (Riglar et al, 2017; Courbet et al, 2015; Archer et al, 2012; McKay et al, 2018) are now available. A challenge is to get sufficient resolution both in terms of detection limits and of abundance of the reporter strains. Nevertheless, continuous improvements are moving these tools from animal and in vitro systems towards the clinics, offering non-invasive monitoring of microbiota functions along the gastrointestinal tract in different disease states (Tanna et al, 2021).

Finally, bacterial isolation, cultivation and characterization—the "original" method for studying microbiotas—has recently experienced a resurgence (Clavel et al, 2017; Afrizal et al, 2022). Anaerobic culturomics approaches have led to the isolation of many strains previously thought to be unculturable. Furthermore, combining these approaches with genomics allows for the prediction of auxotrophies and interdependencies, further increasing our ability to isolate and cultivate strains that may be of high functional relevance (Brugiroux et al, 2017; Kumar et al, 2021). While this is still a slower process than sequencing, the quality of the insights generated is typically very high, as individual strains can be cultivated alone or in combinations and subsequently, genetic, metabolic and non-metabolic interactions studied. Generation of gnotobiotic models that represent different stages of life and disease states is an exciting and rapidly evolving area with tremendous potential to improve our understanding of microbiome functions generally (Brugiroux et al, 2017; Lubin et al, 2023).

We therefore have an ever-increasing range of powerful tools to measure and analyse microbiota composition and function, including spatial and temporal variations. The next frontier, which we urgently need to cross to achieve rational microbiota engineering, is to have high-throughput methods to understand the mechanisms by which microbiota functions drive alterations in host health. Combining organoid biology, mammalian genetics and the skills of organic chemists offers promise (Ahn et al, 2023), however, translating findings from microscale assays in tissue culture to the complexity of a whole, living human being remains a challenge.

## Linking microbiomes to mechanisms of disease

An international effort is currently underway to apply the above tools to unravel causation from correlation in microbiome data (Proctor et al, 2019). Hypothesis-driven bottom-up research approaches have also started to uncover plausible mechanisms.

The immune system contributes to controlling the microbiota through innate and adaptive mechanisms. Excessive or insufficient surveillance has been reported to result in gut pathogen overgrowth (Zheng et al, 2020). Among the first colonizers, the *Bifidobacteriaceae* have been proposed to contribute to the establishment of systemic immune tolerance to the microbiome. Mechanistically, this has been linked to fermentation products generated from HMOs (Jordan et al, 2022; Hitch et al, 2022; O'Neill et al, 2017). In contrast, uncontrolled overgrowth of *Enterobacteriaceae* is associated with Necrotizing Enterocolitis (NEC) (Tarracchini et al, 2021; Gopalakrishna et al, 2019; Olm et al, 2019), a high-mortality condition affecting up to 12% of infants with a birthweight of <1500 g.

In the healthy adult colon microbiota, the abundant products of anaerobic fermentation by obligate anaerobes are SCFAs, including butyrate (Louis and Flint, 2017). Epithelial cells use butyrate in oxidative phosphorylation, which consumes oxygen, keeping the gut lumen hypoxic. Hypoxia favours the growth of obligate anaerobes and limits the growth of facultative aerobes (often opportunistic pathogens such as *Enterobacteriaceae*, but also *Enterococcus* species that are associated with inflammatory disease), closing a virtuous cycle (Rivera-Chávez et al, 2016). Butyrate is used both as an energy source and as a signal to support the development of regulatory T cells in the gut. High levels of

butyrate-producing bacteria correlate with some immunological states (Roduit et al, 2018; Chang et al, 2014; Haak et al, 2018). Other compounds, such as tryptophan metabolites, can also be directly sensed by the gut epithelium to modulate both epithelial biology and immune cell activity (Pham et al, 2014; Li, 2023; Hou et al, 2023). This is certainly only the tip of the iceberg of metabolic interactions. For example tuft cells (a specialized chemosensory epithelial cell type) are perfectly located in the gut epithelium and packed with small molecule-sensing systems (O'Leary et al, 2019).

A major function of the microbiota that intersects with metabolism is the generation of colonization resistance against opportunistic pathogens, via direct competition for resources, or via bacterial warfare (Pickard and Núñez, 2019; Sassone-Corsi and Raffatellu, 2015; Herzog et al, 2023). The complexity of the microbial community in the large intestine ensures that a wide range of metabolic functions effectively exhaust all resources that pathogens could use under the given environmental conditions (Faber et al, 2016; Rivera-Chávez et al, 2016; Stecher, 2021; Brugiroux et al, 2017; Spragge et al, 2023). Suppression of microbiome activity by broad-spectrum antibiotic treatments, chemotherapy, or major diet shifts enable the bloom of organisms such as *Clostridiodes difficile, Enterococcus* spp., and extraintestinal pathogenic *E. coli* (ExPEC) (Stein-Thoeringer et al, 2019; Tavadze et al, 2014; Khanna and Voth, 2023; Worby et al, 2022).

There are also some cases where specific negative functions of microbiota members have been postulated. DNA-damaging agents, including colibactin, may contribute to carcinogenesis (Grasso and Frisan, 2015). *Fusobacterium nucleatum* clade 2 (Fna C2) has been negatively associated with immunotherapy of large intestinal tumours, due to immune-modulation (McCoy et al, 2013; Kostic et al, 2012; Rubinstein et al, 2013). Further species, such as *Enterococcus gallinarum*, have been associated with liver invasion and the exacerbation of autoimmunity (Manfredo Vieira et al, 2018). Nevertheless, the molecular links between specific bacterial species and host phenotype are generally not well understood. The same can be said for inflammatory bowel disease (IBD). It is not clear if pathogenic bacteria or specific functions are drivers of IBD, but some species such as *Klebsiella pneumoniae*, do exacerbate the symptoms and contribute to the chronicity of the disorders (Federici et al, 2022a).

Although we are far from a complete understanding of microbiota function, it is already possible to identify some health-associated properties of the microbiota that could be therapeutically targeted. These include fermentation of dietary fibres, production of micronutrients, modification of bile acids, and competition with enteric pathogens. Prominent negative functions include production of carcinogenic secondary metabolites, and extraintestinal infections. These can act as starting points to develop proof-of-concepts for microbiota engineering.

# Engineering microbiomes

Once we know what is wrong in a microbiome, then ideally, we would like to be able to precisely fix this error, returning everything to a state of health. Humans have been modifying our gut microbiomes for thousands of years through nutrition. Already in pre-history, development of fermentation, as a way of preserving food and enhancing taste and texture, exposed humans to many of the bacterial strains we today consider to be probiotics. More recently, Metchnikoff, and his enthusiasm for lactic acid bacteria as promoters of healthy aging, brought the probiotic concept into the mainstream (Podolsky, 2012; Ezepchuk and Kolybo, 2016). Access to fruit, vegetables and whole grains as reliable sources of dietary fibres has also been a mainstay of healthy eating advice for decades, and we now know supports SCFA production by the microbiota. Broad availability of heavily processed convenience foods in "westernized" settings, combined with high levels of hygiene, have also affected microbiome composition (Sonnenburg et al, 2016). We will start by considering the empirical approaches to microbiota engineering and then explore progress towards precise interventions that could be considered rational at the bacterial species or molecular level.

## Historical approaches to microbiome engineering

**Prebiotics** include a wide range of macromolecules, usually of plant origin, that are non-digestible by the mammalian host and selectively stimulate growth of classes of gut microbes (Fig. 2A). A typical example is resistant starch. This is a complex carbohydrate, insoluble and resistant to degradation by α-amylase. On entering the large intestine, it is a preferred substrate for some butyrate-producing group XIVa *Clostridia*. This metabolic conversion can alter the composition profile of the gut microbiome (Maier et al, 2017) and has proved effective in facilitating weight loss in human studies (Li et al, 2024). Chitin-glucan, as well as soluble dietary fibres like inulin, pectin and cellulose have also shown measurable effects on metabolism and microbiome composition (Ranaivo et al, 2022; Gurry et al, 2018; Ďásková et al, 2023; Wu et al, 2017). Fructo- and galacto-oligosaccharides were suggested to restore bacterial composition in the ageing gut (Arnold et al, 2021), to have anti-obesity effects in rats (Kong et al, 2022), to reduce inflammation in cancer patients (Garcia-Peris et al, 2016) and to promote osteoblastogenesis ex vivo (De Bruyn et al, 2024).

While plant-derived prebiotics dominate the field, milk oligosaccharides can also be classified as a type of prebiotic, and offer specialized benefits for the infant gut health and development. These short glycans specifically support *Bifidobacteria* colonization in human infants (Neumann et al, 2023) and in piglets (Li et al, 2014). Recent data support using HMOs to expand *Bifidobacteria* also in adults (Jacobs et al, 2023).

Probiotics are live microorganisms usually orally delivered, which have associated health benefits (Fig. 2B). These encompass a wide range of species, including, *Lactobacilli* and *Bifidobacteria*, with most commercial products currently derived from fermented foods. Systematic reviews of clinical trials suggest a mild protective effect against antibiotics-associated diarrhoea (Liao et al, 2021). In addition, probiotics are routinely given to premature babies and have been associated with a lower incidence of necrotizing enterocolitis, although it should be noted that their introduction coincided with other changes in routine care so the link may be only correlative (Robertson et al, 2020; Alcon-Giner et al, 2020). Recently, we are seeing a shift to development of gut-derived species of probiotics. *Akkermansia muciniphila* has been associated with improved insulin sensitivity and weight loss (Dao et al, 2016) as well as influencing cancer immunotherapy in epithelial tumours (Routy et al, 2018). Of note, as we move to probiotic strains with a

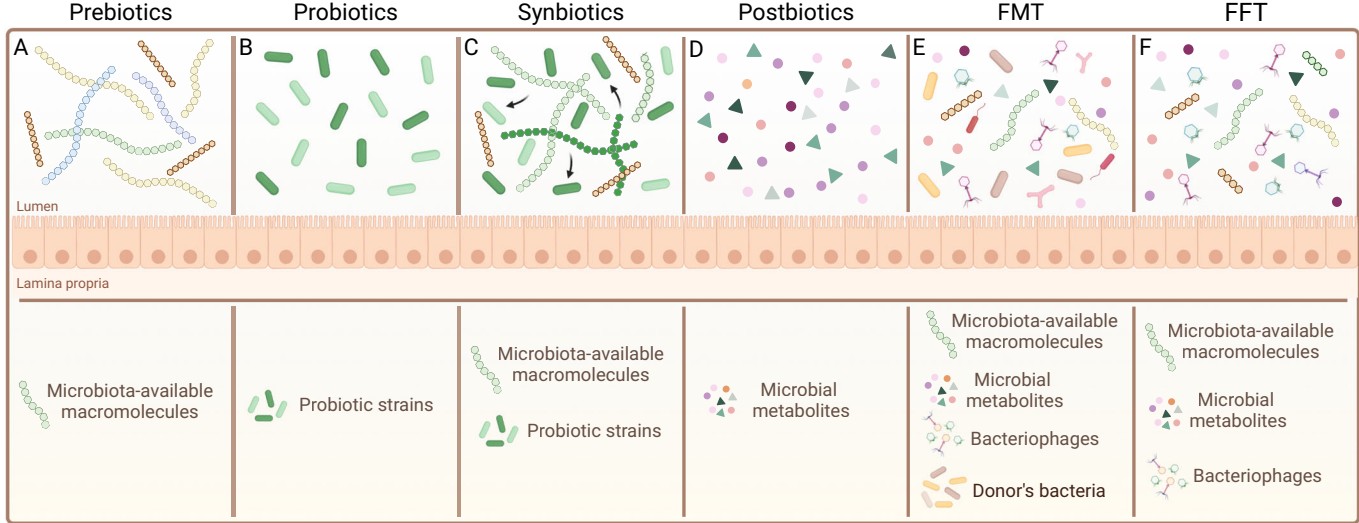

**Figure 2. Existing strategies to manipulate microbiome composition and function.**

(A) Prebiotics are typically indigestible carbohydrate polymers that are known to feed specific classes of beneficial microbes. (B) Probiotics are live bacteria typically delivered orally with the aim of improving microbiota function. (C) Synbiotics are matched combinations of pre- and probiotics, such that the prebiotic should promote probiotic growth and engraftment in the gut. (D) Postbiotics are microbial metabolites that can be delivered orally to mimic the beneficial effects of microbes without live organisms. (E) Fecal Microbiota Transplantation (FMT) is the transfer of total fecal matter from a healthy donor into the gastrointestinal tract of a recipient. (F) Fecal Filtrate Transfer (FFT) is the transfer of filtered fecal supernatant (containing metabolites, bacteriophages, other small macromolecules) to the gastrointestinal tract of a recipient. Graphics were created with BioRender.com.

stronger potential for long-term gut colonization, the associated risks also increase, particularly in immunocompromised patients (Yelin et al, 2019; Salminen et al, 2004; Tóth et al, 2021).

Synbiotic is the term given to a combination of probiotics and prebiotics, designed to increase engraftment (Fig. 2C). In the ideal situation, the prebiotic provides a private nutrient source to the probiotic strain. In very low birthweight neonates, the use of probiotic combined with prebiotics can partially recapitulate the benefits of exclusive breastfeeding (Phavichitr et al, 2021). Paniraghi et al., demonstrated how the use of *L. plantarum* combined with fructo-oligosaccharides supressed sepsis (Panigrahi et al, 2017). In hepatic conditions and obesity, probiotics including *Lactobacilli* and *Bifidobacteria* combined with omega-3-fatty acids have shown ameliorative effects in human subjects (Kobyliak et al, 2017). However, it is crucial to acknowledge that the impact of synbiotics, just like probiotics, can vary significantly across different contexts and individual health conditions.

Postbiotics are a more recent development, which try to deliver the benefits of a probiotic via direct administration of only its bioactive metabolites (Fig. 2D). The most studied examples are SCFAs, including butyrate (Morrison and Preston, 2016). Microbial-derived proteins have also been tested as postbiotics, for example Plovier et al, (Plovier et al, 2017) identified a purified membrane protein from *A. muciniphila* that showed promising results in animal models, as well as in obese and diabetic patients. A challenge remains in delivery to the correct site of action, as SCFAs will be absorbed already in the small intestine upon oral delivery, and bacterial proteins may be digested before arrival at their site of action.

Fecal microbiota transplantation (FMT), involving the transfer of stool from a healthy donor into the gastrointestinal tract of a recipient, has been highly effective in treating recurrent *C. difficile*

infections (Drekonja et al, 2015) (Fig. 2E). As *C. difficile* infections typically follow treatment with broad spectrum antibiotics, a massive open niche for gut bacterial colonization is present that can be closed by fecal exposure. Interestingly, attempts to apply FMT in inflammatory bowel disease, obesity, and metabolic syndrome have been much less successful (Borody and Khoruts, 2012), potentially as these conditions may require displacement of a disease-associated bacterial consortium, rather than simply filling up empty niches. Trials have been reported for the treatment of symptoms of autism spectrum disorder (Kang et al, 2017), steatohepatitis (Zhou et al, 2017), acute graft-vs-host disease (Qi et al, 2018) and cancer immunotherapy (Routy et al, 2018).

Despite its benefits, FMT faces various challenges, including the risk of transferring unwanted pathogens and the identification of suitable "healthy" donors (Danne et al, 2021). The undefined bacterial composition of feces poses a regulatory challenge and renders the technique too dangerous to use in immunocompromised patients. A derivative approach of FMT is known as Fecal Filtrate Transplantation (FFT), i.e., the transfer of a filtered component from the stool of a healthy donor, which includes microbial metabolites, bacteriophages, soluble factors like proteins, cellular debris and oligonucleotides, but excludes live bacteria (Fig. 2F). FFT showed promising outcomes against necrotizing enterocolitis (Brunse et al, 2022) and *C. difficile* infections (Ott et al, 2017), however systematic comparisons between FMT and FFT are essential to fully understand the benefits and limitations of each approach.

While all of these approaches have shown some efficacy, they remain empirical treatments. We rarely know who will respond, which bacteria will engraft in which recipient or what the mechanistic explanation is for any observed positive effect on health. To implement a more robust approach, we need to apply

our functional knowledge of the microbiome to develop rational interventions.

## Designing bacteria and consortia with desired functions

This simplest step from empirical to rational microbiome engineering is in replacing FMT with defined microbes or microbial consortia. This approach allows individual strains to be tested for safety, drug resistance and tolerance, facilitating the generation of licensable defined treatments and enabling the engineering of consortia that cover required functions. Probiotic strains of *E. coli* have been used to limit *Salmonella* Typhimurium (*S.* Tm) infections, by competing for resources under aerobic metabolism (Wotzka et al, 2019; Liou et al, 2022; Arkhammar et al, 1986). In a similar manner, oral administration of *Clostridium butyricum* can enhance the colonization resistance against *C. difficile* by upregulating pathogen-specific intestinal IgA and modulating succinic acid production (Hagihara et al, 2021). Several studies show the benefit of using rationally assembled microbial communities (Brugiroux et al, 2017; Spragge et al, 2023). The pioneering Schaedler flora, developed in the mid-1960s, is believed not to contain any strains of bacteria capable of breaking down urea into ammonia and carbon dioxide. Although it was not designed for this purpose, this feature made it an interesting consortium for treating conditions associated with hyperammonaemia (Shen et al, 2015). Modified versions of this consortium, such as the Altered Schaedler Flora (ASF), have also been used in semi-rational approaches to modulate the gut-brain axis (Muller et al, 2020) or to promote tissue remodelling (Jiménez-Saiz et al, 2020). Additionally, a rationally assembled consortium of strains that should completely degrade simple carbon sources has been assembled and shown to be safe in both mouse models and in human settings (Kurt et al, 2023).

A major application of rationally assembled consortia is to improve or restore colonization resistance against opportunistic pathogens such as *C. difficile*, pathogenic *E. coli* and *Salmonella*. Critically, not all communities provide colonization resistance against all pathogens. Generally, the higher the complexity of the microbiota, the better the protection (Cheng et al, 2022) (for example, Oligo-MM12 and Oligo-MM19 have shown to be able to prevent *Salmonella* colonization to different extents (Brugiroux et al, 2017)). The concept of "nutrient blocking" plays a central role in this process and can act as a designing principle for effective consortia, that consume all accessible nutrient sources for the pathogen (Spragge et al, 2023). Nevertheless, a single carbon-source which is not consumed by a competitor can be sufficient to allow pathogen blooming (Eberl et al, 2021; Gül et al, 2023). There is considerable industrial interest in this field, often with a focus on developing appropriate microbial consortia to treat different conditions, ranging from *C. difficile* infection to graft-versus-host disease.

Genetically engineered bacterial live therapeutics can be designed to do much more than simply compete. While design and initial generation can be challenging, the final strains should be easy to manufacture. Intestinal localization allows targeted therapeutic delivery, enabling gene expression to be regulated in a spatially and temporally controlled manner. Sensors for temperature, pH, oxygen or the presence or absence of specific metabolites or molecules have been developed, and in some cases, they even allow disease-specific regulation (Zou et al, 2023). For

example, Zou and coworkers developed an *E. coli* Nissle strain expressing both a base editing system and a cystatin in a dose-dependent response to the inflammation marker thiosulfate (Zou et al, 2023) (Fig. 3A).

Bacteria have been engineered to encode the required biosynthetic enzymes for a range of small molecules. Promising examples tested in mice include butyrate- (Wang et al, 2022), β-hydroxybutyrate- (Yan et al, 2021) and N-acyl-phosphatidylethanolamine- (Dosoky et al, 2018) producing live therapeutics. These treatments not only lower disease severity, but often also alter microbiota composition (Wang et al, 2022). Similarly, bacteria expressing therapeutic proteins are being developed. Expression of GM-CSF and nanobodies targeting PD-L1 and CTLA-4 in cells programmed for quorum-sensing-induced lysis was shown to be beneficial in a mouse model of colon cancer (Castagliuolo et al, 2005). To overcome the need for bacterial lysis, cargo can be programmed via secretion signals to be actively secreted or delivered in vesicles (Yue et al, 2022).

Bacteria can also be engineered to neutralize or consume toxic compounds internally. For instance, *E. coli* Nissle expressing metallothionein incorporated cadmium and enhanced its removal via feces in poisoned mice (Zou et al, 2022). Similarly, to reduce the ammonia burden that accumulates to toxic levels during hyperammonemia, Kurtz et al. designed an *E. coli* Nissle strain which overproduced arginine, using ammonium as a substrate (Kurtz et al, 2019). Replacement of problematic strains with engineered mutants free of the responsible functions is another therapeutic strategy (Devlin et al, 2016).

Toxin delivery via engineered probiotics offers a designable system to enhance target strain clearance (Tanna et al, 2021). In a gut infection model, the probiotic *E. coli* Nissle was equipped with a sense-and-kill system to release the anti-*Pseudomonas aeruginosa* toxin pyocin S5. In both mice and *Caenorhabditis elegans*, the engineered strain autonomously sensed and reduced the *Pseudomonas aeruginosa* load. Similar strain-specific delivery of toxins has been developed to exclude antibiotic-resistant *Vibrio* spp. (López-Igual et al, 2019). While these approaches are promising for targeted pathogen clearance, we still need to address the risk that the pathogen develops resistance against the toxic cargo or that other microbiota species are harmed upon toxin release (Fig. 3B).

Genetic modification in situ: If the bacteria intended for functional modification are not culturable, or consist of multiple species within the resident microbiota, genetic manipulation may be achieved in situ, i.e., in the intestine, by horizontal gene transfer with an engineered delivery chassis (Dorado-Morales et al, 2023) (Fig. 3C). Seminal work by Ronda et al., successfully targeted about 5% of resident bacteria in conventionally raised mice 6 hours after gavage with *E. coli* donor libraries (Ronda et al, 2019). However, transferred plasmids were mostly lost after 3 days, whereas sequences inserted by transposons were more stable (Ronda et al, 2019). To improve recipient targeting specificity and efficiency, Robledo and coworkers expressed nanobodies on the surface of donor cells, which bound to target-cell-specific antigens and thus strengthened donor-recipient interactions (Robledo et al, 2022). Recently, Brödel et al., employed genetically engineered phage particles carrying base editor genes and specifically modified up to 93% of target bacteria in the mouse intestine upon oral delivery. Efficient adsorption to target cells was reached by optimizing the sequences of phage tail proteins. Furthermore, the phage DNA was

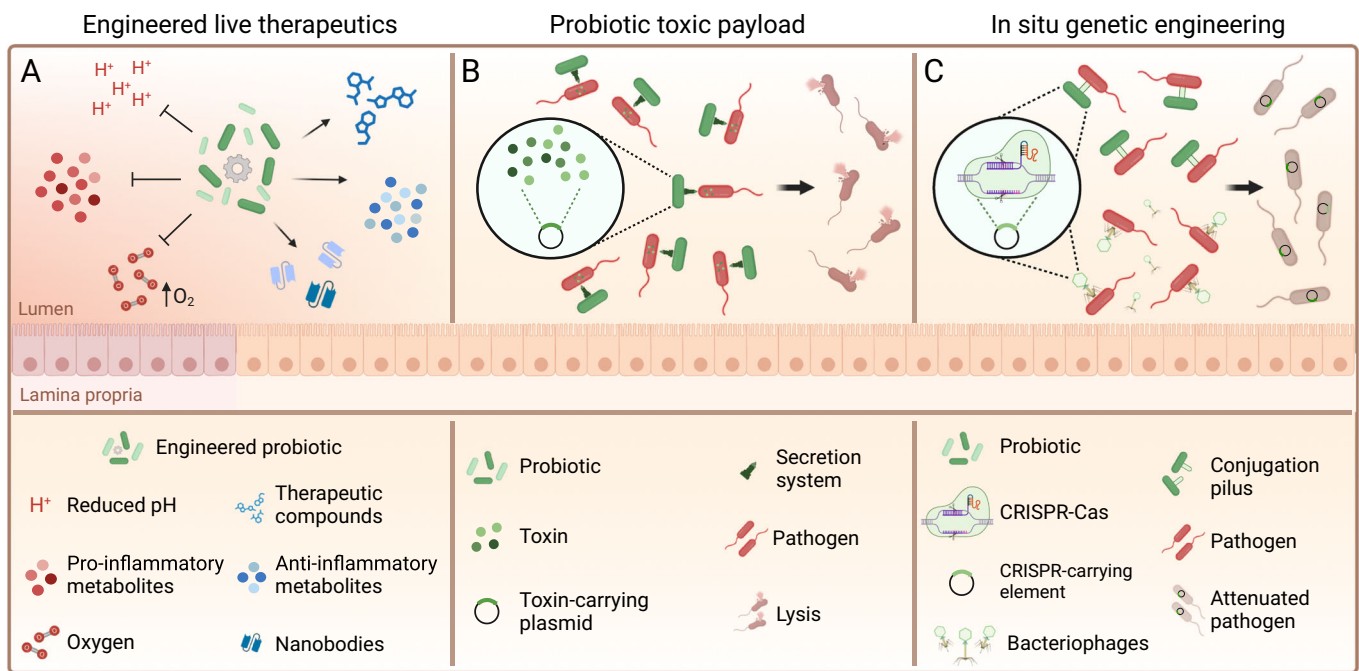

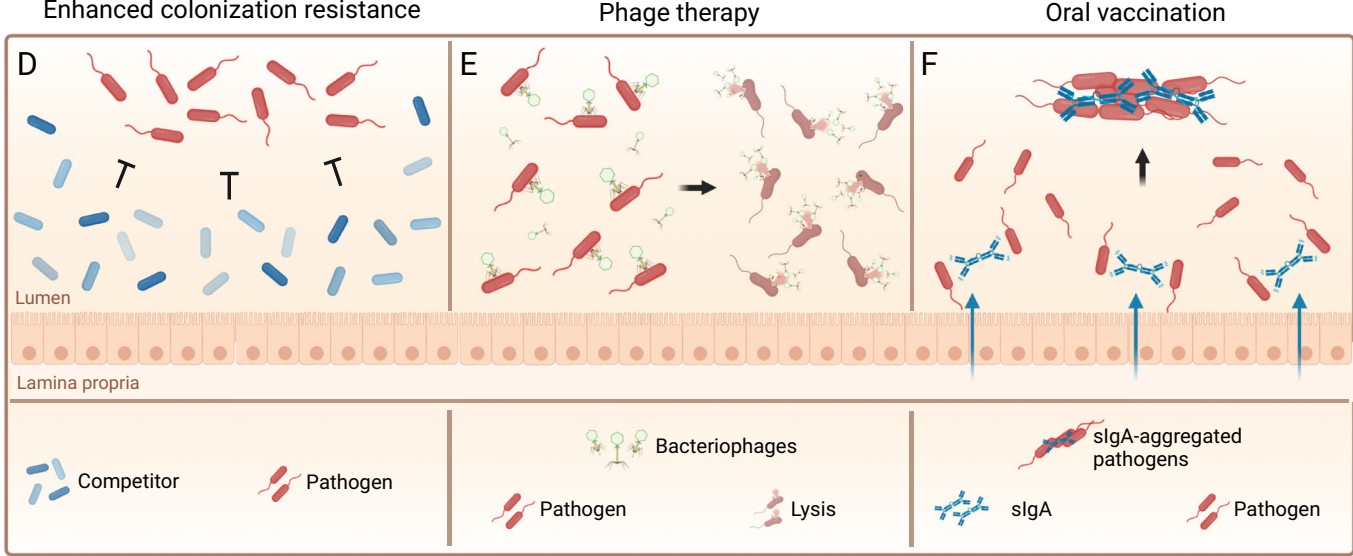

**Figure 3.  Rational microbiome engineering approaches currently under investigation.**

(A) Engineered live therapeutics typically involves a genetically engineered probiotic strain that carries sensor modules and response genes, delivering therapeutic molecules to specific locations in the GI tract. (B) Probiotic toxic payloads can be genetically engineered into safe probiotic strains to kill and clear unwanted species at specific sites in the GI tract. (C) For genetic engineering of strains that cannot be easily cultivated in vitro, or of large numbers of strains simultaneously, approaches have been developed based on broad-host-range plasmids or bacteriophages, to carry out precise genome modifications in situ in the gut lumen. (D) Colonization resistance to a specific pathogen can be enhanced via and understanding of the molecular mechanisms of growth used by the pathogen, and by rationally supplementing strains that compete for carbon, nitrogen or electron acceptors with the pathogenic strains. (E) Bacteriophage (phage) therapy makes use of the high specificity of bacteriophages for individual bacterial strains to suppress colonization of single unwanted members of the microbiome. (F) Oral vaccination induces potent secretory immunoglobulin A responses that aggregate targeted bacteria, while promoting their clearance. Graphics were created with BioRender.com.

modified to prevent replication and packaging in the recipient cells, thereby avoiding uncontrolled spreading (Brödel et al, 2024).

Chassis strains: A critical decision in this type of microbiome engineering is the choice of chassis strain. The availability of

established engineering techniques is key in the choice, and the menu is continuously expanding (Jin et al, 2022). One barrier is the uptake of foreign DNA by phage transduction, natural transformation, electroporation or conjugation. No single method works

Examples of how gut microbiome composition and behaviour is influenced by the host:

(1) Gut anatomy: For example, the rapid constriction of the gut lumen diameter during inflammation (Tropini, 2021; McCallum and Tropini, 2024).

(2) Control of the gut environment:

   (a) Intestinal secretions regulate factors such as pH, the availability of detergent-like molecules, osmolality, nitrogen and carbon sources in the gut lumen (Tropini et al, 2018; Cremer et al, 2017; Coyte et al, 2015).

   (b) Gut content flow rates determine how long microbes can remain in the gut (dwell-time), influencing how many times they can replicate between contractions. Peristaltic mixing also limits population structure formation, affecting phenomena such as crossfeeding. (Tropini, 2021).

   (c) Oxygen depletion by the host prevents aerobic respiration in the gut and favours colonization by beneficial commensals carrying out fermentation (Rivera-Chávez et al, 2016).

(3) Behavioural control: Eating behaviour determines exogenous nutrient influx into the gut, while physical movement typically promotes intestinal motility and decreases transit time. By modulating these two behaviours, major effects on microbiome composition can be rapidly achieved.

(4) Immunological control: Innate mechanisms, such as antimicrobial peptides, are typically active against broad groups of bacteria, and influence the overall density of colonization close to host tissues. Adaptive immunity, on the other hand, can target bacteria down to the strain level, and along the full gastrointestinal tract.

In addition, the microbes themselves govern consortia formation. Negative effects can be exerted via the production of secreted antimicrobials, contact-dependent killing mechanisms and the presence of virulent bacteriophages. Positive effects can come from lysogenic bacteriophages carrying beneficial morons, as well as interactions via environment modification or nutrient exchange.

universally, and although protocols have been optimized for several model organisms, efficiencies remain suboptimal for others. Once inside the target cell, the introduced DNA has to evade degradation by defence mechanisms, including restriction-methylation systems. Removal of restriction sites (Johnston et al, 2019) or their protection by methylation with the appropriate methyl transferases (Zhang et al, 2012) can profoundly increase the efficiency of generating recombinant strains. Additionally, the use of chemical restriction enzymes inhibitors can further improve electroporation output. Anti-defence genes appear to be a common feature in the leading regions of conjugative and mobilizable plasmids (Samuel and Burstein, 2023), and they could be explored to enhance successful DNA transfer of genetically engineered sequences. To be retained in the engineered chassis population, the transformed DNA must replicate as the cells divide. This can be achieved either with plasmids containing their own replication systems, or by insertion of the introduced sequences into the resident chromosome. An array of techniques are available to facilitate the latter, including suicide plasmids for site-specific double crossover homologous recombination, transposons, integrases, or the recently discovered CRISPR-guided transposase systems (Gelsinger et al, 2024), which combine the benefits of transposase insertion efficiency with programmable insertion site specificity. Most commonly, selection for successfully transduced or transformed bacterial cells relies on antibiotic-resistance genes. While this is a valid method for research and development of bacterial live

therapeutics, it raises concerns due to their possible exacerbation of the antimicrobial resistance threat. Several techniques exist to excise resistance markers in an additional recombination step. Alternative selection methods include complementation of auxotrophy or selection based on introduced phenotypes or reporter genes. Finally, successful transcription and translation of the introduced genes requires appropriate promoters and ribosome binding sites, tailored for the chassis organism, with attention to expression levels, and eventual conditional expression requirements (Mimee et al, 2015). Functionality of heterologous gene expression should always be verified. Correct polypeptide folding may require the assistance of chaperones, while post-translational modifications, cofactors or protein interaction partners may be essential for the intended activity.

## Getting in: using gut ecology to reliably introduce beneficial strains or consortia into pre-existing microbiomes

In fact, designing microbes with specific functions is only part of the story in microbiome engineering. The other critical aspect is introducing these species into intact microbiomes, and simply ingesting the bacteria of interest is often unreliable. Here it is critical to understand that a microbiome is not just a collection of microbes, but a functional ecosystem, and introducing a new species requires either an open niche or the ability to displace a species from an existing niche (Coyte et al, 2015). As engineered microbes are often less fit than wild types and most gut microbiomes are difficult to invade, this ability is often lacking. Therefore, we need methods to selectively and reliably engraft engineered strains. A good starting point is to understand the mechanisms naturally at play, including control of the gut environment and nutrient profile, gut immunity, and microbe-microbe interactions (See Box 1) (Arnoldini et al, 2018; Shoaie et al, 2015; Nakajima et al, 2018; Federici et al, 2022b).

It is useful to consider bacterial growth and clearance rates when examining microbiota dynamics. The net-growth rate is the sum of these two factors: increased growth and decrease clearance will result in population expansion, while the opposite will result in population contraction. Changes may be transient, or may converge to a new set-point with different abundances. Bacterial growth is influenced by nutrient availability and environmental factors, while clearance is affected by killing and toxicity mechanisms, as well as gut content flow and bacterial adhesion to intestinal tissues. Healthy gut microbiomes are remarkably stable from day to day, suggesting a balance between growth and clearance, at least over a 24-hour period. Growth and clearance rates likely vary across bacterial populations, due to environmental variation within the gut, and temporal fluctuations (Mark Kim et al, 2020; D'Souza et al, 2021; Ackermann et al, 2008; Freed et al, 2008). Nevertheless, using net growth rates as a guide can inform the rational design of interventions to alter composition or function (or both) of the gut microbiome.

Promoting growth of a strain of interest has been beautifully leveraged by Shepherd et al. by inserting a rare gene cluster sufficient for the utilization of porphyran, a polysaccharide indigestible for most microbes, into a gut *Bacteroides* strain (Shepherd et al, 2018). When porphyran was added to the diet, the engineered *Bacteroides* gained access to a private carbon source, allowing it to colonize to high densities. By providing a private

carbon source, the ability of the strain to generate biomass increases. As long as the clearance rate remains constant, the population size of the strain of interest is expected to grow until an equilibrium is reached or until the private nutrient source is removed. Conversely, as discussed above in the context of designed bacterial consortia, nutrient blocking severely limits the target strain's ability to grow (Fig. 3D).

Other approaches specifically target the bacterial clearance rate. This has been investigated using bacterial warfare (e.g., probiotics engineered to deliver specific toxins), bacteriophages and host immunity.

Bacteriophages (phages) often have a very limited host range (Clokie et al, 2011; Koskella and Meaden, 2013), making them promising agents for highly specific bacterial strain removal (Cepko et al, 2020). Phages that target common human gut bacteria are highly abundant in the environment and can easily be isolated from sewage water against cultivable bacteria (Clokie et al, 2011; Balleste et al, 2022). Oral administration of a single phage strain or of a phage cocktail has been shown to significantly reduce the intestinal load of enteropathogenic *E. coli* without affecting the rest of the microbiome (Galtier et al, 2017; Cepko et al, 2020). Similar approaches have shown some success in suppressing colonization with Vancomycin-resistant *Enterococcus faecalis* (Cheng et al, 2017) and *Klebsiella* (Federici et al, 2022a) with additional microbiota-restoring effects (Fig. 3E). As with probiotics, the functionality of phages can be improved by genetic engineering. Programmable CRISPR-Cas systems allow narrowing the lytic activity even further by selective killing of strains that carry not only the phage receptor, but also undesired genes. This has been used to selectively reduce the intestinal load of unwanted *E. coli* in a mouse gut infection model (Gencay et al, 2024; Lam et al, 2021).

Notably, while phage therapy has had isolated cases of success in the clinics, there have been far more failures, which can largely be traced to (1) the rapid emergence of phage resistance in the target bacterium and (2) population dynamics effects whereby the density of phage and hosts limit interaction and therefore prevent complete host clearance. Rapid evolution of phage resistance can actually be used to our advantage in a process called "phage steering" (Gurney et al, 2020b; Barber et al, 2021; Nang et al, 2024). Phages that attach to an undesired trait of the bacteria, such as a virulence factor (fimbriae, lipopolysaccharide (LPS), pili) or an efflux pump that confers antibiotic resistance, can be used to generate a specific selective pressure. When the bacteria evolve resistance to such phage, which commonly happens by loss or mutation of the phage-binding receptor, the arising mutant population has undergone an evolutionary trade-off and is less fit (Gurney et al, 2020a; Hockenberry et al, 2023). While this does not eliminate the bacteria from the gut, the remaining population may no longer be able to cause disease or may become susceptible to other interventions such as vaccination or antibiotics.

Oral vaccination: The only adaptive immune component present and functional in the gut lumen are secretory antibodies (SIgA in mice, SIgA1 and 2 and SIgM in humans). These antibodies are induced by mucosal infection or oral vaccination and can be specific down to the bacterial strain level (Hockenberry et al, 2023) (Fig. 3F). The functional consequences of IgA targeting remain incompletely explored across the whole microbiome, largely because we have only very limited information on the molecular structures present on the surface of most microbiome members, and on their growth characteristics. In addition,

both canonical (i.e. via the antibody complementarity-determining regions) and non-canonical (i.e. via constant regions and glycosylations of the antibody) binding can influence bacterial population dynamics (Hockenberry et al, 2023; Pabst and Slack, 2020). Nevertheless, we and others have demonstrated the ability of oral-vaccine induced secretory IgA to suppress colonization with pathogenic *E. coli* and *Salmonella* (Lentsch et al, 2022; Diard et al, 2021). Mechanistically, this can be explained by an increased clearance from the gut lumen, due to aggregation and flow in gut content (Moor et al, 2017). Conversely, IgA with measurable affinities for *Bacteroides* capsular polysaccharides increased colonization of colonic mucus by capsule-producing strains (Donaldson et al, 2018). This indicates excellent potential for IgA to modulate intestinal engraftment, but also highlights that using this as a tool requires detailed knowledge of the strain and antigens being targeted, and consequences of IgA interactions.

Combined approaches: While these individual interventions can generate measurable changes in the microbiome and suppression of the strain of interest, the effects are typically incomplete. Far larger effects can be achieved by combining a negative influence on the growth rate with an intervention that increases the clearance rate. Lentsch et al demonstrated the benefit of combining oral vaccination with a probiotic niche competitor to exclude *S.* Tm and non-encapsulated *E. coli* from the gut (Lentsch et al, 2022). Either a competitor alone, or oral vaccination alone, are sufficient to decrease the size of the gut luminal *Salmonella* population around 1000-fold. Both combined can suppress the population more than $10^9$-fold and generate sterilizing immunity (Fig. 4A). In another study, anti-colorectal cancer therapeutics were covalently linked to lytic phages that kill pro-tumoral *F. nucleatum* and combined with prebiotic supplement to promote production of short-chain fatty acids to achieve three goals: reduction of the pathogen and cancer growth as well as promoting restoration of a healthy microbiome and supporting the mucosal immune system (Zheng et al, 2019) (Fig. 4B).

It should be noted that combining strategies to remove specific strains from the gut microbiota requires a deep understanding of the host-microbe interactions involved as well as of the evolutionary trajectories of the microbes upon external pressure, such as host immune responses or phage treatments (Slack and Diard, 2022). Generating IgA responses using whole cell inactivated oral vaccines is simple and straightforward, but some bacterial strains can rapidly change their surface antigenicity (van der Woude and Bäumler, 2004; Mostowy and Holt, 2018) or produce non-immunogenic surface structures such as the *E. coli* K1 capsule (Sande and Whitfield, 2021). In addition, selecting the ideal niche competitor (or cocktail of competitors) requires knowing the metabolism of the target strain, the resident microbiome and the potential competitor pool—an understanding that remains incomplete at the time of writing (Coyte et al, 2021). These strategies show great promise in improving the control of bacterial infections, potentially including eradication of antibiotic-resistant pathogen pools carried in the gut of animals and humans. Nevertheless, there is more work to do before such approaches reach clinical application.

While rational strain replacement often focuses on removing problematic bacteria, the same principles can be applied to introduce beneficial strains by displacing existing microbiome strains occupying the same ecological niche. IgA, bacteriophages and toxins can be used to modulate bacterial clearance rates, while nutrient availability and

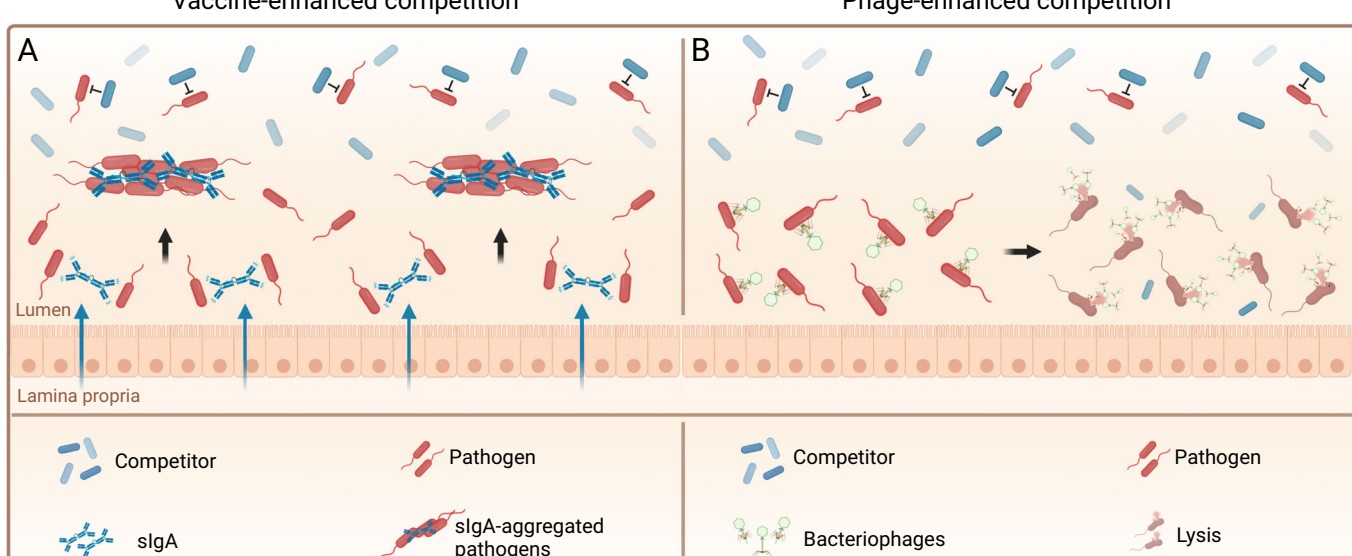

**Figure 4. Combined strategies in microbiome engineering.**

**(A)** Vaccine-enhanced competition. Oral vaccination is combined with carefully selected niche competitor strains to exclude pathogens. While IgA targeting increases the clearance rate of the unwanted strain, niche competition decreases access to nutrients and competes for the available niche, leading to complete elimination. **(B)** Phage-enhanced competition. In this approach bacteriophages are used to induce a competitive disadvantage of the targeted strain and combined with a niche competitor that takes over the available metabolic niche. Graphics were created with BioRender.com.

the gut environment can be used to manipulate growth rates. Integrating research on colonization resistance with advances in probiotic engineering holds great promise.

## Safety

Bacteria, phages, and their genes are biological entities that can multiply, evolve and potentially spread to unintended habitats or genomes, respectively. Stringent containment measures are therefore an essential requirement for any candidate live microbial therapeutic (Huang et al, 2022). The use of auxotrophic strains is a common method for containment (Rovner et al, 2015; Hoffmann et al, 2023). These strains require a specific molecule for survival, which can be provided during therapy but is absent in the environment, rendering escapers nonviable. As an alterative approach, kill-switches are activated by signals that are prevalent in the environment but absent where the activity of the live therapeutic is desired (Stirling et al, 2017; Chan et al, 2016; Knudsen and Karlstrom, 1991; Stirling et al, 2020; Rottinghaus et al, 2022). Bacteria leaving this niche trigger the expression of a toxin, causing them to undergo cell death. Nevertheless, there is a challenge to make these strategies evolutionarily stable. Mutations can inactivate toxin genes of kill-switches, and auxotrophies may be repaired via recombination with horizontally transferred genes. To prevent escape, multiple containment strategies should be implemented in each live therapeutic. Furthermore, therapeutic strains should carry as few antibiotic resistances as possible, such that pathology arising from loss of containment can be treated and the risk of exacerbating the antimicrobial resistance crisis is minimized. A handful of strains have already been tested in human clinical phase 1 and 2 trials with generally encouraging results

concerning safety and tolerability, carrying only minor adverse side effects (Braat et al, 2006). While evidence for efficacy was not found in all cases, some studies are reassuring. Perreault and coworkers tested an *E. coli* Nissle strain, equipped with various safety features and the ability to catabolize methionine (Perreault et al, 2024). High levels of methionine lead to an accumulation of homocysteine in patients, which in turn is implicated in various diseases, particularly, driving pathology in homocystinuria patients. This live-engineered therapeutic was generally well tolerated, with adverse effects independent of dosage. Importantly, metabolic activity was dose-dependent, lowering plasma methionine by approximately one-quarter, only 24 hours after an oral methionine challenge in the highest-dosed cohort. With first proofs of both safety and biological activity in the human body, the future for bacterial live therapeutics looks encouraging.

Another potential aspect of concern is that poorly designed interventions may result in permanent, health-detrimental changes to microbiome composition. Our understanding of the impact of major microbiome shifts remains woefully incomplete. On the other hand, we know that *Enterobacteriaceae* strains turnover every few months in healthy humans without obvious health effects (Martinson et al, 2019), so interventions that seek to replace a potentially pathogenic *E. coli* with a different non-pathogenic strain are likely low risk. Given the extent of our knowledge gaps, this remains a genuine concern and must be carefully monitored in preclinical and clinical trials.

## Conclusions

Although much remains to be learned, this is a field full of optimism. There is considerable interest from the general public

and from industry in improving microbiome function. New approaches to bacterial genetic editing that removes the need for antibiotic resistance genes in selection, as well as wide availability of synthetic DNA and protein engineering have opened up new frontiers. Technological progress, combined with artificial intelligence-based approaches should allow high-throughput analysis of microbiome functions. Our understanding of how the microbiome contributes to a range of diseases is progressing rapidly. As this understanding progresses, we must keep on track with developing the necessary tools to "fix" microbiomes, with potential to influence health worldwide.

## Pending issues

The field of rational gut microbiome engineering has made significant leaps forward in recent years. The increased recognition that we need to combine an understanding of health-supporting microbiome functions, with a thorough knowledge of gut microbial ecology. Nevertheless, major hurdles remain.

The largest challenge of gut microbiome engineering is the extent to which this will need to be personalized. While the microbiome is relatively conserved and stable at the phylum level, it is clear that gut microbiome composition varies extensively from individual to individual (Wade and Hall, 2020). Given the time required to sequence and understand a microbiome, to genetically engineer probiotic strains, and to design, generate and test oral vaccines, any process that requires matching down to the strain level is therefore likely to fail in clinical practice. A key obstacle is therefore to develop interventions that are robust across multiple microbiome configurations. Identifying and targeting common strains and bacterial surface antigens, and using controlled nutritional interventions is showing promise, but still will need evaluation across geographically diverse populations. As a prominent example, the porphyran utilization gene cluster is not rare in populations who regularly consume seaweed as part of their diet (Hehemann et al, 2010), making porphyran unsuitable as a private carbon source for introduced microbes in these populations.

An additional challenge is the genetic and functional diversity of microbes that could potentially be targets for microbiome engineering (Lozupone et al, 2012). The vast majority of these organisms are poorly characterized or have never been cultivated in vitro (Clavel et al, 2017). An extensive amount of work remains in characterising individual species contributions, and to understand whether the function-species relationship is conserved across individuals. Very abundant horizontal gene transfer in the gut clearly adds challenges (Moura de Sousa et al, 2023). Moreover, some functions may be emergent properties of consortia and will depend not only on the presence or absence of a single species but also on the density of colonization (Brunner and Chia, 2019). While developing techniques based on well-characterized bacteria allows us to target and remove opportunistic pathogens from microbiotas, it remains a major gap to address complex functions. To bridge this gap, we need major advances not only in understanding microbiome functions that influence health and the microbes generating these functions, but also in how to design microbiome engineering strategies for more complex targets, with poorly characterized metabolic capacities.

Finally, while our focus here has been on the gut, the biogeography of the microbiome is far more complex (McCallum

and Tropini, 2024; Harris-Tryon and Grice, 2022; Drigot and Clark, 2024). Microbes can be exchanged between body sites, including not only different parts of the gastrointestinal tract, but also skin, oral cavity, respiratory tract and urogenital tracts (Worby et al, 2022). It remains highly plausible that a rational intervention at one site may have unpredictable effects on the microbiome at a distant body site. Our understanding of these exchanges are still limited, with most knowledge coming from transmission of opportunistic pathogens.

## Peer review information

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

## Acknowledgements

ES acknowledges the support of Swiss National Science Foundation (40B2-0_180953, 310030_185128, 310030_219212), and European Research Council Consolidator Grant (865730). This work was supported as a part of NCCR Microbiomes, a National Centre of Competence in Research, funded by the Swiss National Science Foundation (grant number 180575) and mTorus, a project from LOOP Zürich. MD is supported by a SNF professorship (PP00PP_176954), ERC grant ECOSTRAT – 101002643 and Gebert Rüf Microbials (PhagoVax GRS-093/20). ES and MD acknowledge funding provided by the Botnar Research Centre for Child Health as part of the Multi-Investigator Project: Microbiota Engineering for Child Health.

## Author contributions

**Elisa Cappio Barazzone**: Conceptualization; Visualization; Writing—original draft; Writing—review and editing. **Médéric Diard**: Conceptualization; Validation; Writing—original draft; Writing—review and editing. **Isabelle Hug**: Conceptualization; Visualization; Writing—original draft; Writing—review and editing. **Louise Larsson**: Conceptualization; Visualization; Writing—original draft; Writing—review and editing. **Emma Slack**: Conceptualization; Visualization; Writing—original draft; Writing—review and editing.

## Disclosure and competing interests statement

European patent applications EP19177251 and EP22186078 are held by ES and MD. There are no other disclosures or competing financial interests.

