## [Peer Review File · EMBO Molecular Medicine]

Diagnosing and engineering gut microbiomes

Elisa Cappio Barazzone, Médéric Diard, Isabelle Hug, Louise Larsson, and Emma Slack

Corresponding author(s): Emma Slack (emma.slack@hest.ethz.ch)

Review Timeline:

Submission Date:	22nd May 24
Editorial Decision:	11th Jul 24
Revision Received:	5th Sep 24
Editorial Decision:	6th Sep 24
Revision Received:	18th Sep 24
Accepted:	19th Sep 24

Editor: Zeljko Durdevic

Transaction Report:

11th Jul 2024

Dear Prof. Slack,

Thank you for the submission of your manuscript to EMBO Molecular Medicine. We have now heard back from the three referees who agreed to evaluate your manuscript. As you will see from the reports below, the referees are overall positive about its timeliness and interest, but also raise some criticism that should be addressed in a major revision.

I would like to ask you to also amend the following:

- 1) Add corresponding author information on the title page.
- 2) Add up to 5 keywords.
- 3) Remove figures from the main manuscript file and place the legends after references. Please provide detailed description of the figure in the figure legends for figures 2, 3 and 4. Please consider comments of referee #2 regarding figure 2.
- 4) Add "Disclosure and competing interests statement". We updated our journal's competing interests policy in January 2022 and request authors to consider both actual and perceived competing interests. Please review the policy <https://www.embopress.org/competing-interests> and update your competing interests if necessary.
- 5) Glossary: The glossary is meant to explain some of the terms used for laymen. Could you please identify terms that may need an "explanation"?
- 6) Pending issues: At the end of each article is a box highlighting issues that still need further studies and where research efforts should converge. Could you identify some pending issues?

I hope that the referees' comments do not prove too problematic to address and I look forward to reading your next version.

Yours sincerely,

Zeljko Durdevic

*** IMPORTANT INFORMATION ***

- 1) a .doc formatted version of the manuscript text (including Figure legends and tables)
- 2) Separate figure files
- 3) a letter INCLUDING the reviewer's reports and your detailed responses to their comments.

Also, and to save some time should your paper be accepted, please read below for additional information regarding some features of our research articles:

- 1) Glossary: EMBO Molecular Medicine articles will be accompanied by a glossary explaining some of the terms used for laymen. I identified the following:

_____, _____, _____

Could you please help us in identifying terms that may need an "explanation" other terms that we can add to the glossary.

2) For more information: This is a short list of related web links for further consultation by the readers. Could you identify some relevant ones? Examples are patient associations, OMIM related links, databases, authors websites, etc.

3) Pending issues: At the end of each article we will have a box highlighting issues that still need further studies and where research efforts should converge (we call this the Pending issues box). From my reading I would say:

but I can see there may be many more. Could you work on this as well?

4) Disclosure and competing interest statement: Please include a statement declaring any competing commercial interests in relation to your submitted work.

5) Please note that we now mandate that all corresponding authors list an ORCID digital identifier. This takes <90 seconds to complete. We encourage all authors to supply an ORCID identifier, which will be linked to their name for unambiguous name identification.

Currently, our records indicate that the ORCID for your account is 0000-0002-2473-1145.

Link Not Available

-

Thank you,

Zeljko Durdevic

***** Reviewer's comments *****

Referee #1 (Remarks for Author):

##specific comments

page 2 (line 69) - I would avoid the use of long parenthesis. it makes difficult to follow an already complex paragraph. Perhaps the content of the parenthesis could be added as an independent sentence.

page 3 - I agree with the 3 points. I wonder though if some mention of complexities associated with extrachromosomal elements, horizontal gene transfer and genome assembly might be worth mentioning.

page 10 - The section 'Designing bacteria and consortia with desired functions' compiles interesting approaches. It would be interesting to include or discuss the approaches for direct microbiome modifications (ie <https://www.biorxiv.org/content/10.1101/2022.09.30.509847v1>)

page 15 - In this safety section may benefit from discussing briefly the biocontainment strategies developed for genetically modified strains (auxotrophies, kill switches)

Referee #2 (Remarks for Author):

Comment on "Diagnosing and engineering microbiomes"

Analyzing the human microbiome and its influence on health and disease is gaining increasing interest and importance in the scientific community. Individual successes have been achieved, but many questions remain unanswered and the reliability of the

application varies from person to person.

In the first part of this review article, the authors deal with questions about determining the microbiome, the methods and their advantages and imitations, and the link between the microbiome and disease. The second part explains how the microbiome can be modified e.g. by prebiotics, probiotics, synbiotics, FMT and FFT. The third part deals with microbiome-engineering strategies with selected probiotic bacteria or bacteriophages to specifically suppress undesirable pathogenic bacteria. As the microbiome interacts closely with the immune system - indeed, our immune system determines which bacteria are tolerated - methods of strengthening the immune system are also discussed. Lastly, security aspects are addressed and what promising future paths could be based on current knowledge.

Overall, I read the article with great interest. It gives a good overview of the current state of microbiome research. I found it particularly good that, in addition to the optimism that can be discerned, critical comments are also made, pointing out that our knowledge of the complex microbiome is still very limited and that it is not possible to foresee and assess the inherent risks, such as irreversible interventions.

What could be improved?

It would perhaps be advisable to divide the review into 3 - 4 larger parts. I have already tried to provide a structure with first part, second part.....

The figures should definitely be improved and provided with a detailed legend. e.g. in Fig. 2, I don't recognize any difference between prebiotics and synbiotics. in FFT there shouldn't actually be any bacteria present.

Referee #3 (Remarks for Author):

This is a thoughtful review of the potential for engineering microbes to treat human diseases.

Overall, I found the MS was well written and easy to follow. The figures are excellent.

My main contention is that the paper is almost exclusively focussed on the gut microbiome, with no mention of the significant importance of the pulmonary and dermatological microbiota.

It may not be necessary to review these systems in detail, but at a minimum their existence and importance should be acknowledged, and it should be stated that the focus of the current review is the gut microbiota.

The content on dysbiosis is less original than that concerning engineering, and some pruning of the MS would better emphasize the engineering components.

11th Jul 2024

Dear Prof. Slack,

Thank you for the submission of your manuscript to **EMBO** Molecular Medicine. We have now heard back from the three referees who agreed to evaluate your manuscript. As you will see from the reports below, the referees are overall positive about its timeliness and interest, but also raise some criticism that should be addressed in a major revision.

I would like to ask you to also amend the following:

1) Add corresponding author information on the title page.

Done

2) Add up to 5 keywords.

Added

3) Remove figures from the main manuscript file and place the legends after references.

Please provide detailed description of the figure in the figure legends for figures 2, 3 and 4.

Please consider comments of referee #2 regarding figure 2.

Figures have been amended based on Reviewer 2 comments, and are now submitted as separate files.

4) Add "Disclosure and competing interests statement". We updated our journal's competing interests policy in January 2022 and request authors to consider both actual and perceived competing interests. Please review the policy <https://www.embo.press.org/competing-interests> and update your competing interests if necessary.

Done

5) Glossary: The glossary is meant to explain some of the terms used for laymen. Could you please identify terms that may need an "explanation"?

This has been added at the end of the text.

6) Pending issues: At the end of each article is a box highlighting issues that still need further studies and where research efforts should converge. Could you identify some pending issues?

The final section has been renamed to "pending issues", expanded and separated from the conclusions.

I hope that the referees' comments do not prove too problematic to address and I look forward to reading your next version.

Yours sincerely,

Zeljko Durdevic

Zeljko Durdevic

Editor

EMBO Molecular Medicine

*** IMPORTANT INFORMATION ***

- 1) a .doc formatted version of the manuscript text (including Figure legends and tables)
- 2) Separate figure files
- 3) a letter INCLUDING the reviewer's reports and your detailed responses to their comments.

Also, and to save some time should your paper be accepted, please read below for additional information regarding some features of our research articles:

- 1) Glossary: EMBO Molecular Medicine articles will be accompanied by a glossary explaining some of the terms used for laymen. I identified the following:
Could you please help us in identifying terms that may need an "explanation" other terms that we can add to the glossary.

We have added a suggested glossary to the end of the document.

- 2) For more information: This is a short list of related web links for further consultation by the readers. Could you identify some relevant ones? Examples are patient associations, OMIM related links, databases, authors websites, etc.

We have added this section to the end of the document

- 3) Pending issues: At the end of each article we will have a box highlighting issues that still need further studies and where research efforts should converge (we call this the Pending issues box). From my reading I would say: but I can see there may be many more. Could you work on this as well?

This general section has been expanded to form a separate text box at the end of the article. Issues mentioned very specifically relating to the sub-structure of the review have not been moved as I think they make more sense in their current locations.

- 4) Disclosure and competing interest statement: Please include a statement declaring any

competing commercial interests in relation to your submitted work.

This has been added

5) Please note that we now mandate that all corresponding authors list an ORCID digital identifier. This takes <90 seconds to complete. We encourage all authors to supply an ORCID identifier, which will be linked to their name for unambiguous name identification.

Currently, our records indicate that the ORCID for your account is 0000-0002-2473-1145.

ORCID for each author has been added

Thank you,

Zeljko Durdevic

Zeljko Durdevic

Editor

EMBO Molecular Medicine

***** Reviewer's comments *****

Referee #1 (Remarks for Author):

##specific comments

page 2 (line 69) - I would avoid the use of long parenthesis. it makes difficult to follow an already complex paragraph. Perhaps the content of the parenthesis could be added as an independent sentence.

The parentheses have been removed and the sentences restructured accordingly.

page 3 - I agree with the 3 points. I wonder though if some mention of complexities associated with extrachromosomal elements, horizontal gene transfer and genome assembly might be worth mentioning.

This is an excellent point and has been added here with the following text:

- 4) “ Extrachromosomal DNA may encode critical functions, but is extremely challenging to correctly assemble and assign to a host strain, or may be missed altogether in metagenomic analysis(36). Frequent horizontal gene transfer in densely populated environments such as the gut also poses challenges for linking functions to species(37).”

page 10 - The section 'Designing bacteria and consortia with desired functions' compiles interesting approaches. It would be interesting to include or discuss the approaches for

direct microbiome modifications

(ie<https://www.biorxiv.org/content/10.1101/2022.09.30.509847v1>)

This suggestion has been added using the following text:

“Recently, Brödel et al. employed genetically engineered phage particles that carry base editor genes and specifically modified up to 93% of target bacteria in the mouse intestine upon oral delivery. This efficient adsorption to target cells was reached by optimizing the sequences of phage tail proteins. Furthermore, the phage DNA was modified to disallow replication and packaging in the recipient cells in order to avoid uncontrolled spreading.”

page 15 - In this safety section may benefit from discussing briefly the biocontainment strategies developed for genetically modified strains (auxotrophies, kill switches)

This suggestion has been added, using the following text:

“The use of auxotrophic strains is a common method for containment. These strains require a specific molecule for survival, which can be provided during therapy but is absent in the environment, rendering escapers nonviable. Contrariwise, kill-switches are activated by signals that are prevalent in the environment but absent where the activity of the live therapeutic is desired (in the patient). Escapers are thus induced to express a toxin and commit suicide whenever they leave their intended niche. Another strategy to reduce the risk of a harmful spread of engineered microbes is to employ strains that require laboratory conditions to multiply and are unable to colonize the gut, but rather perform their therapeutic effect while passing through the intestine. Additional benefits of non-colonizers are a better control of their dosage, and that loss of activity due to mutations can be avoided. Mutations can inactivate toxin genes of kill-switches and auxotrophies may be repaired via recombination with horizontally transferred genes. To prevent accidental escapes by mutation, multiple containment strategies should be implemented in each live therapeutic. Furthermore, therapeutic strains should carry as little antibiotic resistances as possible, such that incidences of loss of containment can be controlled with these drugs”

Referee #2 (Remarks for Author):

Comment on "Diagnosing and engineering microbiomes"

Analyzing the human microbiome and its influence on health and disease is gaining increasing interest and importance in the scientific community. Individual successes have been achieved, but many questions remain unanswered and the reliability of the application varies from person to person.

In the first part of this review article, the authors deal with questions about determining the microbiome, the methods and their advantages and imitations, and the link between the microbiome and disease. The second part explains how the microbiome can be modified e.g. by prebiotics, probiotics, synbiotics, FMT and FFT. The third part deals with microbiome-engineering strategies with selected probiotic bacteria or bacteriophages to specifically suppress undesirable pathogenic bacteria. As the microbiome interacts closely with the immune system - indeed, our immune system determines which bacteria are tolerated - methods of strengthening the immune system are also discussed. Lastly, security aspects are addressed and what promising future paths could be based on current knowledge.

Overall, I read the article with great interest. It gives a good overview of the current state of microbiome research. I found it particularly good that, in addition to the optimism that can be discerned, critical comments are also made, pointing out that our knowledge of the complex microbiome is still very limited and that it is not possible to foresee and assess the inherent risks, such as irreversible interventions.

Many thanks for the positive comments

What could be improved?

It would perhaps be advisable to divide the review into 3 - 4 larger parts. I have already tried to provide a structure with first part, second part.....

Thank you for this suggestion – we have added “parts 1-3” to the section titles and numbers the sub-headings to increase the structure.

The figures should definitely be improved and provided with a detailed legend. e.g. in Fig. 2, I don't recognize any difference between prebiotics and synbiotics. in FFT there shouldn't actually be any bacteria present.

Thank you for these suggestions. We have drastically re-worked the figures as suggested. Changes include:

- 1) Increasing the clarity of the time-line in figure 1
- 2) Remove extraneous information from figure 2 and 3 (e.g. endogenous microbiota) and adding clearer keys and definitions of the symbols used.
- 3) Titles added to figure 4, together with removing extraneous information to improve clarity.
- 4) Clear figure-legends added for all figures.

Referee #3 (Remarks for Author):

This is a thoughtful review of the potential for engineering microbes to treat human diseases. Overall, I found the MS was well written and easy to follow. The figures are excellent.

Thank you for the positive comments

My main contention is that the paper is almost exclusively focussed on the gut microbiome, with no mention of the significant importance of the pulmonary and dermatological microbiota.

It may not be necessary to review these systems in detail, but at a minimum their existence and importance should be acknowledged, and it should be stated that the focus of the current review is the gut microbiota.

Thank you for pointing this out. We have now corrected this via the following edits:

The title now reads “**Diagnosing and engineering gut microbiomes**”

Introduction:

“From the moment of birth, mammals inhabit a microbial world. Our first exposure is usually to maternal microbes ingested during birth. The maturing microbiome then rapidly changes during the first years of life, reaching an adult-like composition at around the age of three years in humans¹. All body surfaces become colonized during this phase of life, and the importance of the microbiome on the skin and in the respiratory and urogenital tracts has been recently reviewed elsewhere. For the purpose of this review we will focus on the gut.”

Pending issues:

“Finally, while we have here focused on the gut, the biogeography of the microbiome is far more complex than this. Microbes can be exchanged between body sites, including not only different parts of the gastrointestinal tract, but also skin, oral cavity, respiratory tract and urogenital tracts. It remains highly plausible that a rational intervention at one site may have unpredictable effects on the microbiome at a distant body site. We still understand relatively little about this exchange, with most knowledge coming from transmission of opportunistic pathogens.”

The content on dysbiosis is less original than that concerning engineering, and some pruning of the MS would better emphasize the engineering components.

Thank you for this suggestion – we have gone through and removed the less interesting material and repetition from the dysbiosis section.

6th Sep 2024

Dear Prof. Slack,

Thank you for the submission of your manuscript to EMBO Molecular Medicine. I am pleased to inform you that we will be able to accept your manuscript pending the following final amendments:

- 1) Please upload figures as individual high-resolution files.
- 2) Add section heading "References" and correct the reference citation in the text and reference list. In the text a reference should be cited by author and year of publication. Include a space between a word and the opening parenthesis of the reference that follows. In the reference list, citations should be listed in alphabetical order. Where there are more than 10 authors on a paper, 10 will be listed, followed by "et al.". Also, please remove DOIs. Please check "Author Guidelines" for more information. <https://www.embopress.org/page/journal/17574684/authorguide#referencesformat>
- 3) Please enter all funding information in the "Acknowledgments".
- 4) Remove "textbox" in the heading of "Pending issues".
- 5) Add following sentence to the figure legends when BioRender was used to create the figure: "Graphics were created with BioRender.com."
- 6) Rename "Textbox 1" to "Box 1".
- 7) Remove "Further information" section with links.
- 8) As part of the EMBO Publications transparent editorial process initiative EMBO Molecular Medicine will publish online a Review Process File (RPF) to accompany accepted manuscripts. This file will be published in conjunction with your paper and will include the anonymous referee reports, your point-by-point response and all pertinent correspondence relating to the manuscript. Let us know whether you agree with the publication of the RPF.

I look forward to receiving the revised version of your manuscript.

Yours sincerely,

Zeljko Durdevic

The authors addressed the minor editorial issues.

19th Sep 2024

Dear Prof. Slack,

We are pleased to inform you that your manuscript is accepted for publication and is now being sent to our publisher to be included in the next available issue of EMBO Molecular Medicine.

Your manuscript will be processed for publication by EMBO Press. It will be copy edited and you will receive page proofs prior to publication.

You will soon be contacted by our publisher Springer Nature to sign your publishing license. When you login to the customer service website, please use the token/code copied below to waive the article publication charges. Should you experience any difficulty, please email publishing@embo.org.

Waiver token: MTTYNTC4MTA0NG
